# TNF-α in Combination with Palmitate Enhances IL-8 Production via The MyD88- Independent TLR4 Signaling Pathway: Potential Relevance to Metabolic Inflammation

**DOI:** 10.3390/ijms20174112

**Published:** 2019-08-23

**Authors:** Amal Hasan, Nadeem Akhter, Areej Al-Roub, Reeby Thomas, Shihab Kochumon, Ajit Wilson, Merin Koshy, Ebaa Al-Ozairi, Fahd Al-Mulla, Rasheed Ahmad

**Affiliations:** 1Microbiology & Immunology Department, Dasman Diabetes Institute, Kuwait City 15462, Kuwait; 2Medical Division, Dasman Diabetes Institute, Kuwait City 15462, Kuwait; 3Genetics and Bioinformatics, Dasman Diabetes Institute, Kuwait City 15462, Kuwait

**Keywords:** inflammation, interleukin-8, tumor necrosis factor-alpha, free fatty acids, toll-like receptor 4

## Abstract

Elevated levels of IL-8 (CXCL8) in obesity have been linked with insulin resistance and type 2 diabetes (T2D). The mechanisms that lead to the profound production of IL-8 in obesity remains to be understood. TNF-α and saturated free fatty acids (FFAs) are increased in obese humans and correlate with insulin resistance. Hence, we sought to investigate whether the cooccurrence of TNF-α and FFAs led to increase the production of IL-8 by human monocytes. We found that co-stimulation of human monocytes with palmitate and TNF-α led to increased IL-8 production as compared to those stimulated with palmitate or TNF-α alone. The synergistic production of IL-8 by TNF-α/palmitate was suppressed by neutralizing anti- Toll like receptor 4 (TLR4) antibody and by genetic silencing of TLR4. Both MyD88-deficient and MyD88-competent cells responded comparably to TNF-α/Palmitate. However, TIR-domain-containing adapter-inducing interferon (TRIF) inhibition or interferon regulatory transcription factor 3 (IRF3) knockdown partly blocked the synergistic production of IL-8. Our human data show that increased adipose tissue TNF-α expression correlated positively with IL-8 expression (*r* = 0.49, *P* = 0.001). IL-8 and TNF-α correlated positively with macrophage markers including CD68, CD163 and CD86 in adipose tissue. These findings suggest that the signaling cross-talk between saturated fatty acid palmitate and TNF-α may be a key driver in obesity-associated chronic inflammation via an excessive production of IL-8.

## 1. Introduction

Obesity-mediated low-grade chronic inflammation plays a key role in the development of various diseases including Type 2 diabetes, atherosclerosis, and hepatic steatosis. In obesity, adipose tissue is dysregulated by the infiltration of immune cells, associated with increased production of cytokines and chemokines. In particular, pro-inflammatory macrophage accumulation in adipose tissue is an important feature of obesity, correlating with local expression of inflammatory markers including tumor necrosis factor-α (TNF-α), interleukin-6 (IL-6), interleukin-1β (IL-1β) and C-C Motif Chemokine Ligand 2 (CCL2) [1,2,3,4]. These inflammatory mediators contribute to the infiltration of circulatory monocytes into the adipose tissue which play a central role in the development and maintenance of obesity-associated chronic low-grade inflammation [5,6]. More recently, the chemokine system, particularly interleukin-8 (IL-8; also known as CXCL8), has been recognized as critical in metabolic inflammation [7,8,9]. A recent study demonstrated that individuals with T2D display higher IL-8 levels compared with those with normoglycemia, indicating greater adipose tissue-associated inflammation and worse glycometabolic profile [10].

IL-8 is a proinflammatory cytokine that belongs to the CXC chemokine superfamily [11], and is secreted by several cell types, including monocytes/macrophages, T-cells, adipocytes, endothelial and epidermal cells [12,13]. IL-8 is regarded as a multifunctional chemokine having both chemoattractant and mitogenic effects on neutrophils [14], monocytes, T-cells, vascular endothelial cells, and vascular smooth muscle cells [15,16,17,18]. Moreover, several studies have demonstrated a role for IL-8 in the recruitment and activation of hepatic macrophages in chronic liver disease [19], and in the pathogenesis and progression of non-alcoholic fatty liver disease (NAFLD) [20,21,22,23,24,25,26]. Similarly, IL-8 promotes macrophage infiltration in the adipose tissue [27], inducing local and systemic inflammation, and thus, may represent a link between adipose tissue dysfunction and insulin resistance [20,21,22,23,24,25,26,27,28,29]. Several studies have investigated the potential underlying mechanisms leading to increased IL-8 production in obesity and metabolic disease. In this regard, an association was described between IL-8 and hepatic metabolic pathways in humans. IL-8 was shown to be secreted by hepatocytes when stimulated with free fatty acids (FFAs) through activation of the nuclear factor kappa-light-chain-enhancer of activated B cells (NF-κβ) and c-Jun-N-terminal kinase (JNK) pathways [30]. Moreover, IL-8 enhanced its own expression (IL-8 mRNA) in human adipocytes, and in turn, adipocytes expressed the receptor for IL-8 leading to an autocrine effect on these cells. This inflammatory stimulation creates a vicious circle of IL-8 production in human adipocytes via extracellular signal-regulated kinases (ERK) pathway and/or p38 mitogen-activated protein kinase (MAPK) pathway [28]. Altogether, these findings indicate that IL-8 plays a critical role in both the initiation and maintenance of inflammation in the adipose tissue. Furthermore, in vitro studies demonstrated that IL-8 induces insulin resistance by inhibiting insulin-induced Akt phosphorylation in human adipocytes. Thus, inhibition of IL-8 production and/or action may represent a novel target for the prevention and/or treatment of T2D and associated complications. In addition to its function in the recruitment of inflammatory cells, IL-8 may play a central role in influencing adipocyte physiology.

Despite the presented evidence of IL-8 involvement in metabolic disease, studies specifically designed for assessing the mechanisms involved in this association is lacking [10]. Many factors have been shown to promote obesity-associated inflammation. Obesity and T2D is associated with increased levels of FFAs, which have been reported to contribute in the development of chronic inflammation and insulin resistance [31]. FFAs can activate inflammatory signaling pathways through various mechanisms, one of which is through activation of the innate immune receptor TLR4 [32].

TLR4 is a member of the pattern-recognition receptor family and plays a key role in innate immunity. In response to microbial pathogens and components (such as lipopolysaccharide; LPS) and damaged tissue-associated signals or alarmins, TLR4 activates pro-inflammatory signaling pathways. LPS binding to TLR4 can activate one of two different intracellular signaling pathways, these include the myeloid differentiation factor 88 (MyD88)-dependent pathway, and the MyD88-independent pathway which utilizes the adaptor TRIF. Activation of MyD88 triggers a downstream signaling cascade leading to the activation of NF-κβ and MAPK pathways. Ligand-induced internalization of TRIF leads to its association with TLR4 and IRF3 activation along with its downstream signaling cascade [33]. In the adipose tissue, FFAs enhance the polarization of classically-activated M1 inflammatory macrophages, which secrete proinflammatory cytokines including TNF-α, IL-1β, and IL-6 [34]. Elevated levels of TNF-α are particularly inflammatory with several studies showing a direct correlation between TNF-α and insulin resistance [35,36]. Indeed, deletion of TNF-α or its receptors in obese animals was demonstrated to reduce macrophage infiltration into the adipose tissue, and to improve insulin sensitivity. Several studies have demonstrated consistently higher levels of IL-8, FFAs and TNF-α in the circulation of individuals with obesity/T2D [31,37] and we recently showed that coexistent of TNF-α and saturated fatty acids enhanced CCL2 production [38]. Therefore, we asked whether the co-occurrence of FFAs and TNF-α could trigger higher IL-8 production by monocytes. Herein, we show that long chain fatty acids, particularly palmitate, substantially enhance IL-8 expression in monocytes when co-cultured with TNF-α. We further show that TLR4 is required for this synergistic activity, and that the TRIF/IRF3 (i.e., MyD88-independent) arm of the TLR4 signaling pathway is critical. In addition, our human data show that increased adipose tissue TNF-α expression correlated positively with IL-8 expression. IL-8 and TNF- α correlated positively with macrophage markers including CD68, CD163 and CD86 in adipose tissue.

## 2. Results

### 2.1. IL-8 Production is Markedly Increased by Palmitate/TNF-α in Monocytes

IL-8 gene expression and protein secretion were significantly higher in monocytes cotreated with palmitate and TNF-α compared to cells treated with palmitate or TNF-α alone (Figure 1A,B; Appendix A). Similarly, palmitate greatly enhanced TNF-α-mediated production of IL-8 by THP-1-derived macrophages (Figure 1C,D). In monocytes, elevated IL-8 expression was confirmed by confocal microscopy (green fluorescence; Figure 1E). To determine whether other long chain fatty acids have a similar synergistic effect on TNF-α-mediated production of IL-8, we cotreated monocytes with TNF-α and either oleate, myristate, or stearate. Although all four fatty acids had a synergistic effect on TNF-α-mediated production of IL-8, palmitate induced the strongest response (Figure 2A). Since the anti-inflammatory role of short chain fatty acids (SCFAs) such as butyrate, propionate, and acetate is well documented [39,40], we assessed whether one such SCFA, namely propionate, synergizes with TNF-α for IL-8 production. As expected, cotreatment of monocytes with propionate and TNF-α failed to enhance the production of IL-8 (Figure 2B), thereby demonstrating that SCFAs do not have the same synergistic effect as long chain fatty acids. Altogether, these results indicate that TNF-α-mediated induction of IL-8 is particularly enhanced by palmitate. 

### 2.2. The Synergistic Expression of IL-8 by TNF-α/Palmitate Requires TLR4

Treatment of monocytes with anti-TLR4 neutralizing antibody prior to co-stimulation with palmitate and TNF-α significantly suppressed IL-8 mRNA expression and protein secretion (Figure 3A,B). To verify that the synergistic induction of IL-8, by co-treatment with palmitate and TNF-α, was dependent on TLR4 signaling, we transfected cells with TLR4 small interfering RNA (siRNA), which resulted in approximately 40% reduction in TLR4 mRNA expression levels compared to scrambled siRNA control (Figure 3C). Our data showed that siRNA-mediated knockdown of TLR4 led to the same result (Figure 3D,E), confirming that the synergistic induction of IL-8 by palmitate and TNF-α requires TLR4. In addition, we demonstrated that a well-known ligand for TLR4, namely LPS, had a similar synergistic effect with TNF-α in inducing IL-8 production, which further confirms the role of TLR4 in this synergy. (Figure 3F,G). Moreover, inhibition of mitogen-activated protein kinases (MAPKs) and NF-κβ pathways by specific inhibitors significantly reduced the production of IL-8 in response to TNF-α/palmitate stimulation (Figure 4A,B). NF-κβ and AP-1 are common downstream transcription factors of TLR4 and TNF-αR1. In monocytes, co-stimulation with TNF-α along with either palmitate or LPS induced higher NF-κβ/AP-1 activity compared to stimulation with palmitate, LPS or TNF-α alone (Figure 4C,D) and this NF-κB/AP-1 activity correlated with IL-8 secretion (Figure 4E).

### 2.3. The Synergistic Production of IL-8 by TNF-α/Palmitate does not Depend on MyD88

MyD88 null (−/−) THP-1 monocytes were used to determine whether the synergistic effect of palmitate/TNF-α requires the MyD88-dependent pathway. In the setting of MyD88 deficiency, palmitate alone failed to induce IL-8 production, whereas the synergistic action of palmitate and TNF-α on IL-8 production was not affected (Figure 5A,B). Similarly, NF-κβ/AP-1 activity was not observed after stimulation with palmitate alone, whereas co-stimulation of MyD88−/− monocytes with palmitate and TNF-α induced higher NF-κβ/AP-1 activity compared to stimulation with TNF-α alone (Figure 5C).

### 2.4. Synergistic Production of IL-8 by Palmitate/TNF-α Requires TRIF/IRF3 Signaling Pathway 

Pretreatment of monocytes with chlorpromazine (CPZ), which is an inhibitor of clathrin-dependent endocytosis, followed by co-treatment with palmitate and TNF-α caused a significant suppression in the synergistic production of IL-8, both at the mRNA and protein levels (Figure 6A,B). To confirm the involvement of TRIF in this synergistic action, we used both small molecule (Resveratrol) and peptide-based (pepinh-TRIF) inhibitors. Pretreatment of monocytes with resveratrol abrogated the synergistic effect of palmitate and TNF-α (Figure 6C,D). A similar effect was observed after blocking TLR4-TRIF interaction with pepinh-TRIF (Figure 6E,F). To verify that the synergistic induction of IL-8, by co-treatment of monocytes with palmitate and TNF-α, was dependent on the TLR4/IRF3 signaling axis, we transfected cells with IRF3 siRNA. Our data showed that the expression of IL-8 mRNA as well as protein secretion were significantly reduced in IRF3 siRNA-transfected cells after co-stimulation with palmitate and TNF-α (Figure 7A,B). Since IRF3 is also activated by the TLR3 signaling pathway, we investigated whether the requirement for TLR4 activation could be abrogated by specific activation of TLR3 with polyinosinic-polycytidylic acid (poly I:C) [41]. Our data showed that poly I:C-mediated IRF3 activation induced a synergistic effect with TNF-α resulting in increased IL-8 mRNA expression and protein production (Figure 7C,D). Taken together, our results demonstrated that IRF3 is a key effector of the synergistic actions of palmitate and TNF-α in monocytes.

### 2.5. Association between IL-8, TNF-α and Macrophage Markers in Subcutaneous Adipose Tissue from Humans

To determine whether our in vitro findings are in line with data obtained from human adipose tissue, we sought to investigate the association between IL-8 and TNF-α, as well as macrophage markers, in the adipose tissue of individuals with varying degrees of adiposity. Our data showed a direct correlation between IL-8 mRNA expression and the body mass index (BMI; *r* = 0.38; *P* = 0.01; *n* = 44) (Figure 8A) and percent body fat (PBF) (*r* = 0.39; *P* = 0.016) (Table 1). Consistently, IL-8 mRNA and protein levels were significantly (*P* = 0.001 and *P* = 0.002, respectively) higher in individuals with obesity as compared to normoweight or overweight controls (Figure 8B–D). Similarly, TNF-α mRNA expression was directly correlated with the BMI (*r* = 0.34; *P* = 0.02, *n* = 48) (Figure 9A), and TNF-α mRNA and protein levels were significantly higher (*P* = 0.005 and *P* = 0.007, respectively) in individuals with obesity as compared to those with normoweight or overweight controls (Figure 9B–D). Consistent with our in vitro findings, adipose tissue IL-8 was directly correlated with TNF-α, both at the mRNA (*r* = 0.49; *P* = 0.001; *n* = 42) and protein (*r* = 0.74; *P* < 0.0001) levels (Figure 10A,B). In addition, IL-8 was directly correlated with macrophage markers CD68 (although did not reach statistical significance; *r* = 0.27; *P* = 0.088; *n* = 42), CD86 (*r* = 0.38; *P* = 0.01, *n* = 40) and CD163 (*r* = 0.4; *P* = 0.006, *n* = 43) (Figure 10C–E). Similarly, TNF-α was directly correlated with CD68 (*r* = 0.47; *P* = 0.0007; *n* = 48), CD86 (*r* = 0.47; *P* = 0.001; *n* = 46) and CD163 (*r* = 0.57; *P* < 0.0001; *n* = 47) (Figure 10F–H). Overall, our clinical data confirmed that IL-8 and TNF-α are elevated in individuals with obesity and that these two cytokines are directly and strongly associated with each other and with macrophage markers in human adipose tissue. 

## 3. Discussion 

Pro-inflammatory macrophage recruitment into the adipose tissue is a key driver of the low-grade chronic inflammation that is associated with obesity and insulin resistance [1,2]. IL-8 promotes macrophage infiltration in the adipose tissue [27] thereby inducing local and systemic inflammation, and thus, may represent a link between adipose tissue dysfunction and insulin resistance [20,21,22,23,24,25,26,27,28,29]. A recent study demonstrated that individuals with T2D display higher IL-8 levels than those with normoglycemia, indicating greater adipose tissue-associated inflammation, worse glycometabolic profile and low vitamin D levels [10]. Although previous studies have described some mechanisms by which IL-8 increases adipose dysfunction [7,20,22], the mechanisms that lead to increased IL-8 in obesity needs further exploration, particularly when in the presence of FFAs, which are high in obesity. In this study, we demonstrate that FFAs (palmitate, oleate, myristate, stearate) and TNF-α synergistically enhance IL-8 production by monocytes. This synergetic effect was not observed with the SCFA propionate, confirming the proinflammatory property of FFAs. 

IL-8 is secreted by several cell types, including monocytes/macrophages, T-cells, adipocytes, endothelial and epidermal cells [12,13], in response to different stimuli including TNF-α [42], FFAs [43], and LPS [44]. However, no studies have elucidated the synergistic effect between FFAs and TNF-α on the production of IL-8. It is well known that FFAs can activate TLR4 signaling in monocytes/macrophages and adipocytes [45,46,47,48]. However, whether synergy between FFAs (such as palmitate) and TNF-α involves TLR4-mediated signaling remained to be elucidated. In this study, we provide evidence that the synergistic effect between palmitate and TNF-α is dependent on TLR4. Firstly, production of IL-8, by the synergetic action between palmitate and TNF-α, was suppressed by neutralization of TLR4. Secondly, stimulation of TLR4 deficient cells with palmitate and TNF-α did not support synergistic production of IL-8. Lastly, activation of TLR4 with LPS (a known agonist of TLR4) along with TNF-α also resulted in the synergistic production of IL-8. Similarly, to LPS, palmitate can initiate signals through TLR4 and activate NF-κβ, MAPKs, and AP-1, which are known to play a role in inflammation and insulin resistance [45,48,49,50,51,52,53,54]. In obesity-associated metabolic inflammation, FFAs may act as ligands for TLR4 and thus induce TNF-α production by monocytes/macrophages, which suggests pathological interactions between FFAs and TLR4 [32,55]. 

In mice, endotoxemia causes inflammation via the TLR4-mediated signaling pathway [56]. Our data showed that palmitate mimics the effect of LPS in inducing TNF-α-mediated IL-8 production, and that this effect is mediated via NF-κβ/AP-1 activation. In monocytes, TLR4 and TNF-αR agonistic responses are known to activate NF-κβ/AP-1 classical signaling pathways [48,57,58]; thus, FFAs may initiate inflammatory signaling via activation of TLR4 in monocytes. It is well established that TLR4-mediated signaling activates one of two major downstream signaling pathways, namely the MyD88-dependent and MyD88-independent (TRIF/IRF3-dependent) pathways. MyD88-dependent signaling pathways involve interaction between interleukin-1 receptor-associated kinase 1(IRAK1) and TNF receptor-associated factor (TRAF6), leading to the activation of NF-κβ and MAPKs and AP-1. MyD88-independent (TRIF/IRF3) pathway involves activation of downstream molecules including tank-binding kinase 1 (TBK1) and inhibitor of Iκβ kinase ε (IKKε), which in turn phosphorylate and activate IRF3 [59,60,61]. In our study, we sought to investigate which of the two signaling pathways is involved in the synergistic palmitate and TNF-α induced IL-8 production. We found that synergistic IL-8 production in response to palmitate and TNF-α was not affected by the absence of MyD88, whereas IL-8 production in response to palmitate alone was completely abolished. We and others previously demonstrated that palmitate induced the activation of NF-κβ/AP-1 via the MyD88-dependent pathway leading to the production of inflammatory mediators [48,54,62,63,64]. Here, we further show that palmitate and TNF-α synergistically activate NF-κβ/AP-1 via the MyD88-independent pathway. Accordingly, we speculated that the TRIF/IRF3 signaling pathway downstream of TLR4 may act as a regulator of this synergy. Indeed, the role of TRIF/IRF3 signaling in this synergistic effect between palmitate and TNF-α was confirmed. In this regard, we showed that blockade of ligand-induced TLR4 internalization inhibits synergy between palmitate and TNF-α. These results are in line with previous studies showing that inhibition of clathrin-dependent TLR4 endocytosis abrogates LPS-induced expression of TRIF dependent genes including interferon-β (IFN-β), regulated on activation, normal T cell expressed and secreted (RANTES) and IL-6 [65,66,67]. Collectively, these data suggest that clathrin-dependent endocytosis of TLR4 plays an important role in the expression of TRIF target genes, including the synergistic effect of palmitate and TNF-α on IL-8. Moreover, experiments using inhibitors of TLR4/TRIF complex formation showed that TRIF involvement was critical for the synergy between palmitate and TNF-α for IL-8 production. These findings are in line with previous work demonstrating that blockade of TLR4/TRIF complex formation inhibits LPS-induced cytokine production [68]. 

In addition, we showed that silencing of the *IRF3* gene resulted in reduced IL-8 production in response to the synergistic action of palmitate and TNF-α. Since IRF3 is also activated by the TLR3 signaling pathway, we investigated whether the need for TLR4 activation could be circumvented by specific activation of TLR3 using polyinosinic-polycytidylic acid (poly I:C) [41]. Our data showed that poly I:C-mediated IRF3 activation induced a synergistic effect with palmitate/TNF-α resulting in increased IL-8 production. We recently demonstrated that palmitate leads to phosphorylation of IRF3 [38], like other upstream activators such as LPS [69]. TLR3/TLR4 were previously shown to activate NF-κβ/MAPKs in MyD88 deficient cells [70]. Similarly, IRF3 activation by various receptors along with activation of NF-κβ/MAPKs led to the activation of target genes including IP-10, RANTES, IFN-stimulated gene 56 and arginase II [71]. TRIF activation by LPS/TLR4 signaling led to the activation of IRF3/IRF7 and NF-κβ-dependent signaling pathways [72]. Since MyD88-competent and MyD88-deficient cells showed comparable activity, we suggest that the synergistic action of palmitate and TNF-α is predominantly mediated via the TRIF/ TRIF-related adaptor molecule (TRAM)/IRF3 signaling axis.

The levels of IL-8, TNF-α and FFAs are elevated in the circulation of obese humans and have been shown to orchestrate macrophage accumulation and inflammation in the adipose and other tissues [5,31,36,73]. Therefore, this suggests that the simultaneously elevated plasma levels of these crucial markers may induce significant pathophysiological changes in the adipose tissue compartment in obesity, including changes in insulin sensitivity and lipid metabolism. Similarly, a recent study has demonstrated that IL-8 is increased in the circulation of individuals with T2D, which was associated with worse inflammatory and cardiometabolic profile [10]. Our results from human adipose tissue showed a direct correlation between IL-8 and TNF-α, which were also directly correlated with macrophage markers including CD68, CD163 and CD86. Adipose tissue of individuals with obesity exhibit elevated expression of TNF-α [35], IL-6 [74], CCL2 [75], FFAs [31], and IL-8 [28,37] when compared with normo-weight controls. Plasma IL-8 was found to be increased in individuals with obesity and to be related to adiposity and TNF-α [37]. Taking our findings into consideration, we surmise that the co-presence of TNF-α and FFAs in obesity may enhance IL-8 production, thereby contributing to macrophage recruitment into the adipose tissue and metabolic inflammation. Obesity is associated with increased expression of adipokines including TNF-α, IL-6, and CCL2 which can cause damage to adipocytes and the release of FFAs [2,76], and the combined effect of TNF-α and FFAs may trigger further IL-8 production. In the adipose tissue, both adipocytes and immune cells express receptors for TNF-α and FFAs, and thus, act as the source as well as the targets of proinflammatory signals. 

In conclusion, our results show that the synergistic interaction between palmitate and TNF-α is dependent on the TLR4/IRF3 pathway and leads to increased IL-8 production, thereby providing the pathophysiological link between FFAs, TNF-α, and IL-8.

## 4. Materials and Methods

### 4.1. Materials

Palmitate (Cat#P5585), oleate (cat#O1008), myristate (cat#M3128), Stearate (cat# W303518), sodium propionate (cat#P5436), LPS (cat#L4391), Trolox (cat#238813), NDGA (cat#74540), PMA (cat#P1585), Poly I:C (cat#P9582), chlorpromazine (cat#C8138), and resveratrol (cat#R5010) were purchased from Sigma (San Diego, CA, USA). Recombinant human TNF-α (cat# 210-TA-100) was obtained from R&D systems (Minneapolis, MN, USA). Pepinh-TRIF (trif inhibitory peptide: RQIKIWFQNRRMKWKK-FCEEFQVPGRGELH-NH2; Pepinh-Control: RQIKIWFQNRRMKWKK-SLHGRGDPMEAFII-NH2. cat# tlrl-pitrif), Quanti-blue medium (cat#rep-qb-2) were purchased from InvivoGen (San Diego, CA, USA). Cell lysis buffer (cat#9803) were obtained from Cell Signaling (Cell Signaling Technology Inc., Danvers, Massachusetts, United States). IRF3 (ID 3661) Trilencer-27 Human siRNA (SR320690), TLR4 (ID 7099) Trilencer-27 Human siRNA (SR322051) and scrambled (control) siRNA (SR30004) were purchased from OriGene (OriGene Technologies Inc. Rockville, MD, USA). Antibiotics zeocin (cat#ant-zn), normocin (cat#ant-nr) and hygroGold (cat#ant-hg) were purchased from InvivoGen (San Diego, CA, USA). TLR4 neutralizing antibody (cat#mabg-htlr4), IgA2 isotype control (maba2-ctrl) InvivoGen (San Diego, CA, USA).

### 4.2. Human Cell Lines

Human monocytic leukemia THP-1 cell line was purchased from American Type Culture Collection (ATCC). THP-1-XBlue cells stably expressing a secreted embryonic alkaline phosphatase (SEAP) reporter inducible by NF-κβ and AP-1 as well as THP-1-XBlue™-defMyD cells deficient in MyD88 activity (MyD88−/− THP-1 cells) were purchased from InvivoGen (San Diego, CA, USA).

### 4.3. Participant Data and Samples

A total number of 58 adults (13 normoweight, 18 overweight and 27 obese) were included in the study. All participants had provided written informed consent, and the study was conducted in accordance with the ethical principles of the Declaration of Helsinki and approved (04/07/2010; RA-2010-003) by the Ethical Review Committee of Dasman Diabetes Institute, Kuwait. After measuring participant’s body weight (kg) and height (to the nearest 0.5 cm), body mass index (BMI) were calculated as weight/height^2^ (kg/m^2^) and used as an overall index of adiposity. Peripheral blood was collected after an overnight fast (minimum 8 h), and plasma glucose, serum total cholesterol, triglycerides (TG), and high-density lipoprotein (HDL) cholesterol were measured using the Siemens Dimension RXL chemistry analyzer (Diamond Diagnostics, Holliston, MA, USA). Low-density lipoprotein (LDL) cholesterol was also estimated. Glycated hemoglobin (HbA1c) was determined using Variant™ (Bio-Rad, Hercules, CA, USA). High sensitivity C-reactive protein (hsCRP) was measured using ELISA kit (BioVendor, Asheville, NC, USA). Anthropometric and biochemical parameters are shown in Table 2. In addition, adipose tissue samples (~0.5 g) were collected via a standard surgical procedure from abdominal subcutaneous fat pads situated lateral to the umbilicus [77]. Freshly collected adipose tissues (~50–100 mg) were preserved in RNA later and stored at −80 °C until use. Adipose tissue total RNA was purified using RNeasy kit (Qiagen, Valencia. CA, USA) as per manufacturer’s instructions.

### 4.4. THP-1 Cell Culture and Generation of Macrophages

THP-1 cells (1 × 10^6^ cells/mL) were cultured using 12-well plates (Costar, Corning Incorporated, Corning, NY, USA) in RPMI-1640 medium (Gibco, Life Technologies, Grand Island, NE, USA) containing 10% fetal bovine serum (FBS) (Gibco, Life Technologies, Grand Island, NY, USA), 2 mM glutamine, 1 mM sodium pyruvate, 10 mM HEPES, 100 µg/mL Normocin, 50U/mL penicillin and 50 μg/mL streptomycin and cells were incubated at 37 °C in 5% CO_2_ under humidity. THP-1 cells were differentiated into macrophages by treatment with phorbol 12-myristate 13-acetate (PMA; 10 ng/mL) for 3 days in routine culturing media.

### 4.5. THP-1 Cell Stimulation for IL-8 Measurements 

THP-1 or NF-kB/AP-1 reporter cells were plated at a concentration of 1 × 10^6^ cells/well in 12-well plates and stimulated with palmitate (200μM; Sigma, CA, USA) and/or TNF-α (10 ng/mL) for 24 h at 37 °C. Cells were harvested for RNA isolation and conditioned media were collected and stored at −80 °C until use for determining IL-8 production. IL-8 secreted protein was measured in cell supernatants using Human DuoSet Elisa Kit (R&D systems, Minneapolis, MN, USA) as per manufacturer’s instructions. 

### 4.6. Real-Time Quantitative RT-PCR

Total cellular RNA was extracted using RNeasy Mini Kit (Qiagen, Valencia, CA, USA) and cDNA was synthesized from 1 μg of total RNA using high capacity cDNA reverse transcription kit (Applied Biosystems, Foster city, CA, USA). For real-time PCR, cDNA (50ng) was amplified using TaqMan^®^ Gene Expression MasterMix (Foster City, CA 94404, USA), TaqMan Gene Expression Assay products (Applied Biosystems, Foster city, CA, USA) (GAPDH: Hs03929097_g1; IL-8: Hs00174103_m1; TNF-α: Hs01113624_g1; CD68: Hs02836816_g1; CD86: Hs0157026_mL; TLR4: Hs00152939; CD163: Hs00174705_m1; IRF3: Hs01547283_m1) containing forward and reverse gene-specific primers and a target-specific TaqMan^®^ minor groove binder (MGB) (Applied Biosystems, Foster city, CA, USA) probe labeled with 6-fluorescein amidite dye at the 5′ end and a non-fluorescent quencher MGB at the 3′ end of the probe, using a 7500 Real-Time PCR System (Applied Biosystems, CA, USA). The mRNA levels were normalized against glyceraldehyde 3-phosphate dehydrogenase (GAPDH) mRNA. The comparative C_T_ method was used to analyze the relative gene expression of specific genes using the formula 2^-ΔΔC^_T_ [78] whereby ΔΔC_T_ = [C_T_ gene of interest – C_T_ endogenous control; ΔC_T_] – control sample ΔC_T_ (in the case of participants’ samples, the highest ΔC_T_ obtained from normoweight individuals was used as control). In this regard, the relative mRNA fold expression for each specific gene was calculated by first normalizing the C_T_ values to an endogenous control (GAPDH) (ΔC_T_) and then normalizing the ΔC_T_ of each gene expression to control sample ΔC_T_. 

### 4.7. Immunocytofluorescence

THP-1 cells (1 × 10^6^ cells) were washed in PBS and coated on slides using cytospin technique at 600 rpm for 3 min. The slides were then fixed in 4% formaldehyde for 10 min and washed three times in cold PBS. Cells were permeabilized with 0.5% Triton X-100 in PBS for 10 min, washed three times in cold PBS, blocked in 1% bovine serum albumin (BSA) for 1hr, and then incubated overnight with primary antibody (1:100 dilution, rabbit anti-human IL-8 polyclonal antibody; Abcam^®^ ab106350) at room temperature. Cells were washed three times in PBS/Tris buffer and incubated with secondary antibody (goat anti-rabbit antibody conjugated with Alexa Fluor^®^ 488; Abcam^®^ ab150077) for 1hr. After several washes in PBS, the cells nuclei were counterstained with 4′,6-diamidino- 2-phenylindole (DAPI) and mounted (Vectashield, Vectorlab, H1500). Confocal images were collected using inverted Zeiss LSM710 Spectral Confocal Microscope (Carl Zeiss, Gottingen, Germany) and EC Plan-Neofluar 40 ×/1.30 oil DIC M27 objective lens (Carl Zeiss, Gottingen, Germany). After sample excitation using a 405 nm and 488 nm line of an argon ion laser, optimized emission detection bandwidths were configured using Zeiss Zen 2010 software (https://www.zeiss.com/microscopy/int/products/microscope-software.html).

### 4.8. NF-κβ/AP-1 Activity Measurement 

NF-kB/AP-1 reporter monocytes are THP-1 cells that are stably transfected with a reporter construct, expressing SEAP gene under the control of a promoter inducible by NF-κβ and AP-1 transcription factors. Upon stimulation, NF-κβ and AP-1 are activated and lead to the secretion of SEAP in cell supernatant. THP-1-XBlue cells were cultured in complete RPMI medium containing Zeocin (200 µg/mL) to select for cells expressing the SEAP-NF-κβ/AP-1 reporter. THP-1-XBlue™-defMyD cells were cultured in complete RPMI medium containing Zeocin (200 µg/mL) and HygroGold (100 µg/mL). THP-1 XBlue cells were stimulated with palmitate (200 μM) and TNF-α (10 ng/mL), alone or in combination, for 24 h at 37 °C. SEAP levels were measured in conditioned media after incubation with Quanti-Blue for 4 h, and the medium OD were measured at 650 nm wavelength.

### 4.9. Gene Silencing

Gene silencing was performed using transient transfection method and using Amaxa Cell Line Nucleofector Kit V (Lonza, Cologne, Germany) and Amaxa Electroporation System (Amaxa Inc., Cologne, Germany) as per manufacturer’s instructions. For transient transfection, THP-1 cells (1 × 10^6^) were resuspended in Nucleofector^®^ solution and transfected separately using 30 nM IRF3 siRNA and scrambled negative control siRNA. After 36 h, transfected cells were treated with palmitate (200 μm) and TNF-α (10 ng/mL) for 24 h. Cells were harvested for RNA isolation and conditioned media were collected for measurement of IL-8 production in supernatants. Immunoblotting analysis or real-time RT-PCR was also performed to confirm the effective suppression of constitutive TLR4 or IRF3 expression in THP-1 cells transfected with TLR4 siRNA or IRF3 siRNA and scrambled negative control siRNAs.

### 4.10. Statistical Analysis

The GraphPad Prism software (version 7.04; San Diego, CA, USA) was used for statistical analyses. Data from in vitro experiments are expressed as mean ± SEM values and group means were compared using unpaired *t*-test. Data from human samples were assessed for linear dependence between two variables using the non-parametric Spearman’s *r* test, and the group medians were compared using the non-parametric Kruskal-Wallis test. A *P*-value of less than 0.05 were considered statistically significant.

## Figures and Tables

**Figure 1 ijms-20-04112-f001:**
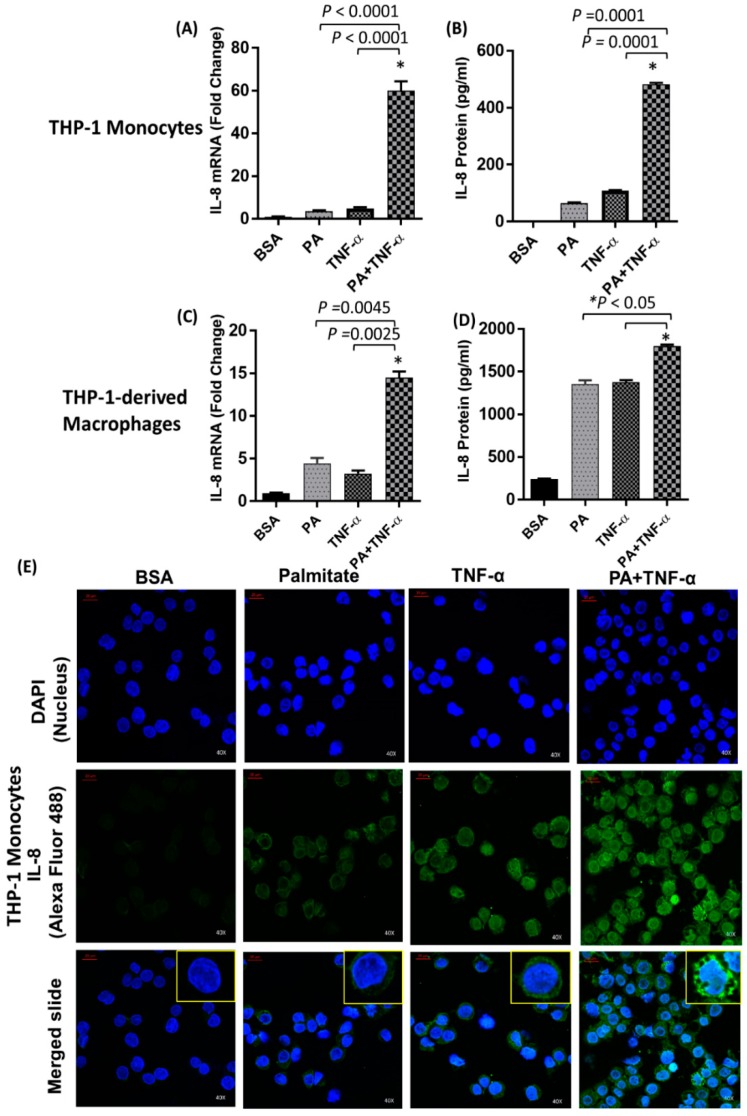
Palmitate and TNF-α synergistically induce IL-8 in THP-1 monocytes. (**A**) THP-1 cells were treated for 24 h with palmitate (PA; 200 µM) alone or in combination with TNFα (10 ng/mL). Total RNA was extracted, and *IL-8* mRNA was quantified by real-time PCR. Relative mRNA expression was expressed as fold change. (**B**) Secreted IL-8 protein in culture media was quantified by ELISA. (**C**,**D**) THP-1-derived macrophages were treated as described above and IL-8 mRNA expression and protein secretion were determined. The results obtained from three replicates of each experiment are shown. Data are expressed as mean ± SEM. (**E**) THP-1 cells were immune-stained for confocal microscopy as described in the Methods. IL-8 expression is shown by green fluorescence (inset) while nuclei are stained blue with DAPI (40X magnification).

**Figure 2 ijms-20-04112-f002:**
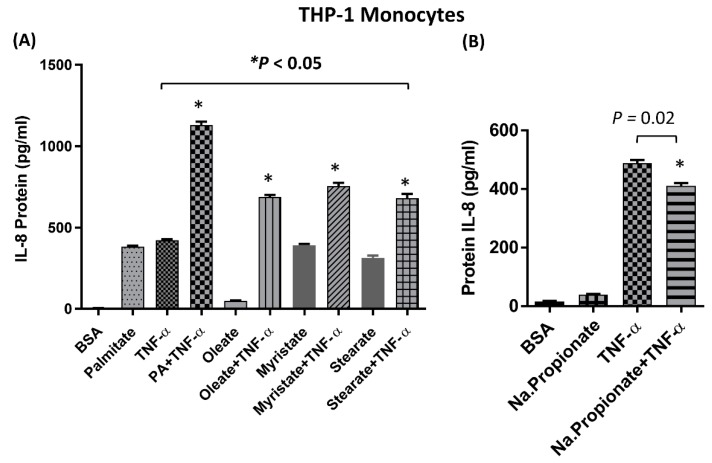
Effect of different fatty acids combined with TNF-α for IL-8 production. (**A**) THP-1 cells were treated for 24 h with different long chain free fatty acids (FFAs) alone or in combination with TNF-α. IL-8 protein was quantified in culture media by ELISA. (**B**) Monocytes were treated with the short chain fatty acid propionate alone or in combination with TNF-α, and IL-8 protein secretion was quantified by ELISA. The results obtained from three replicates of each experiment are shown. Data are expressed as mean ± SEM. * Indicates significant difference.

**Figure 3 ijms-20-04112-f003:**
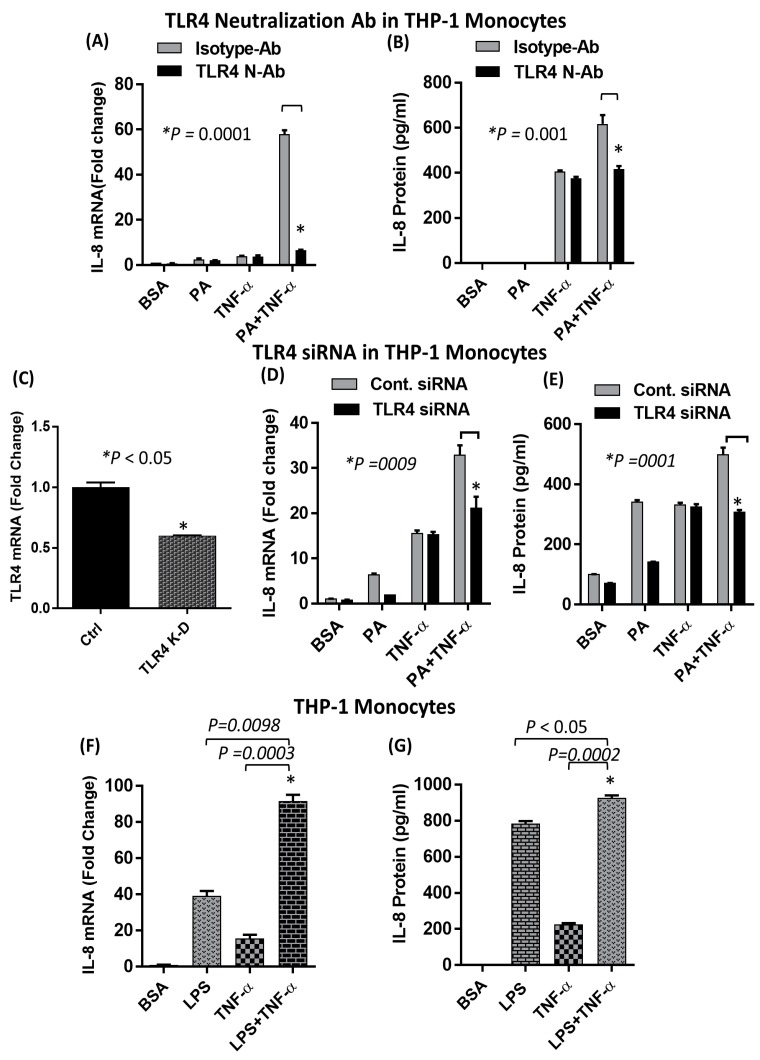
**Interference of TLR4 suppresses the synergistic production of IL-8**. (**A**,**B**) Monocytes were treated with 2 µg/mL of neutralizing TLR4 mAb or isotype-matched control (IgA2) for 40 min. Antibody-treated cells were stimulated with palmitate, TNF-α or a combination of both and incubated for 24 h. *IL-8* mRNA was quantified by real-time PCR and secreted IL-8 protein was quantified in culture media by ELISA. (**C**) Monocytes were transfected with either control or TLR4 siRNA. (**D**,**E**) Monocytes with TLR4-knockdown were treated with palmitate alone or in combination with TNF-α for 24 h. IL-8 expression was determined at both the mRNA and protein levels. (**F**,**G**) Monocytes were treated with lipopolysaccharide (LPS) alone or in combination with TNF-α for 24 h. IL-8 expression was determined at both the mRNA and protein levels. The results obtained from three replicates of each experiment are shown. Data are expressed as mean ± SEM. * Indicates significant difference

**Figure 4 ijms-20-04112-f004:**
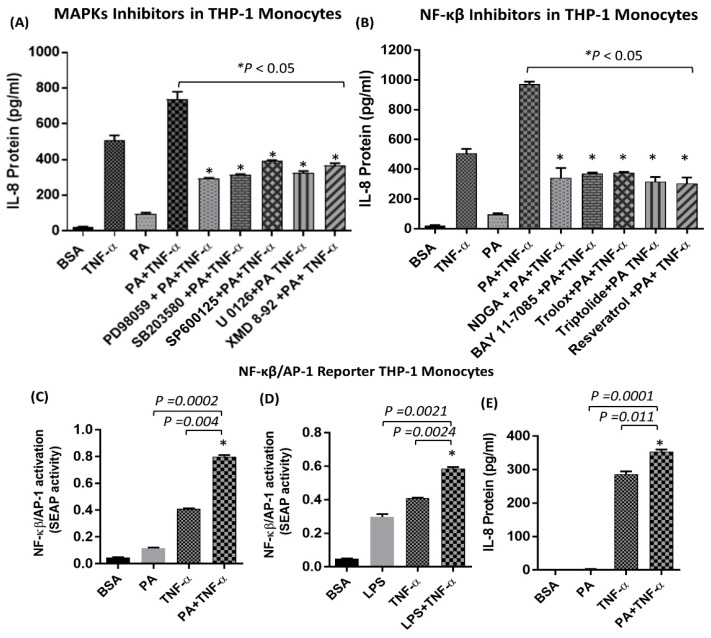
MAPKs and NF-κβ pathways are required for the synergistic production of IL-8 by palmitate/TNF-α. (**A**) Cells were pretreated with mitogen activated protein extracellular signal-regulated kinase (MEK)-ERK inhibitors (U0126: 10 µM; PD98059: 10 µM) or JNK inhibitor (SP600125: 20 µM) or p38 inhibitor (SB203580: 10 µM) or MEK5/ERK5 inhibitor (XMD 8-92: 1 µM) or for 1 h and then treated with PA and TNF-α alone or both for 24 h. Secreted levels of IL-8 protein were quantified in culture supernatants by ELISA. (**B**) Cells were pretreated with NF-κβ inhibitors (Trolox, 10 µM; NDGA, 10 µM; BAY 11-7085, 10 µM; Triptolide, 10µM; Resveratrol, 5µM) for 1 h and then treated as indicated for 24 h. Secreted levels of IL-8 protein were quantified in culture supernatants by ELISA. (**C**–**E**) THP1-XBlue cells (monocytic cells stably expressing a secreted embryonic alkaline phosphatase (SEAP) reporter inducible by NF-κβ and AP-1) were treated with PA and LPS, alone or in combination with TNF-α, for 24 h. Cell culture media were assayed for IL-8 protein as well as SEAP reporter activity (degree of NF-κβ/AP-1 activation). The results obtained from three replicates of each experiment are shown. Data are presented as mean ± SEM. * Indicates significant difference.

**Figure 5 ijms-20-04112-f005:**
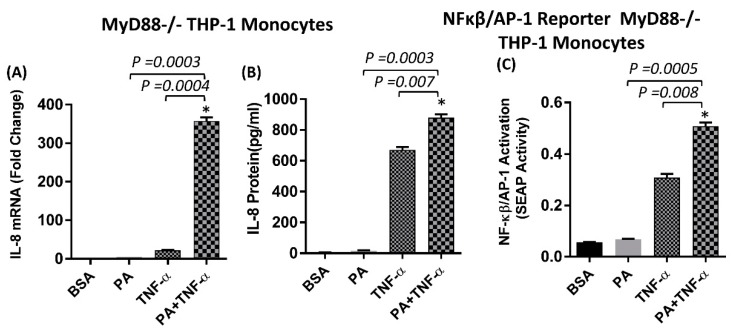
**Synergistic induction of IL-8 by Palmitate/TNF-α involves the MyD88-independent pathway.** (**A**,**B**) Monocytes deficient in MyD88 activity (THP-1 XBlue defMyD cells) were treated with palmitate (200 µm), BSA (Vehicle) or TNF-α (10 ng/mL) alone or in combination for 24 h. IL-8 gene expression was determined by real-time PCR and secreted IL-8 protein was quantified in culture media by ELISA. (**C**) Cell culture media were also assayed for SEAP reporter activity representing the degree of NF-κβ/AP-1 activation. The results obtained from three replicates of each experiment are shown. All data are expressed as mean ± SEM. * Indicates significant difference.

**Figure 6 ijms-20-04112-f006:**
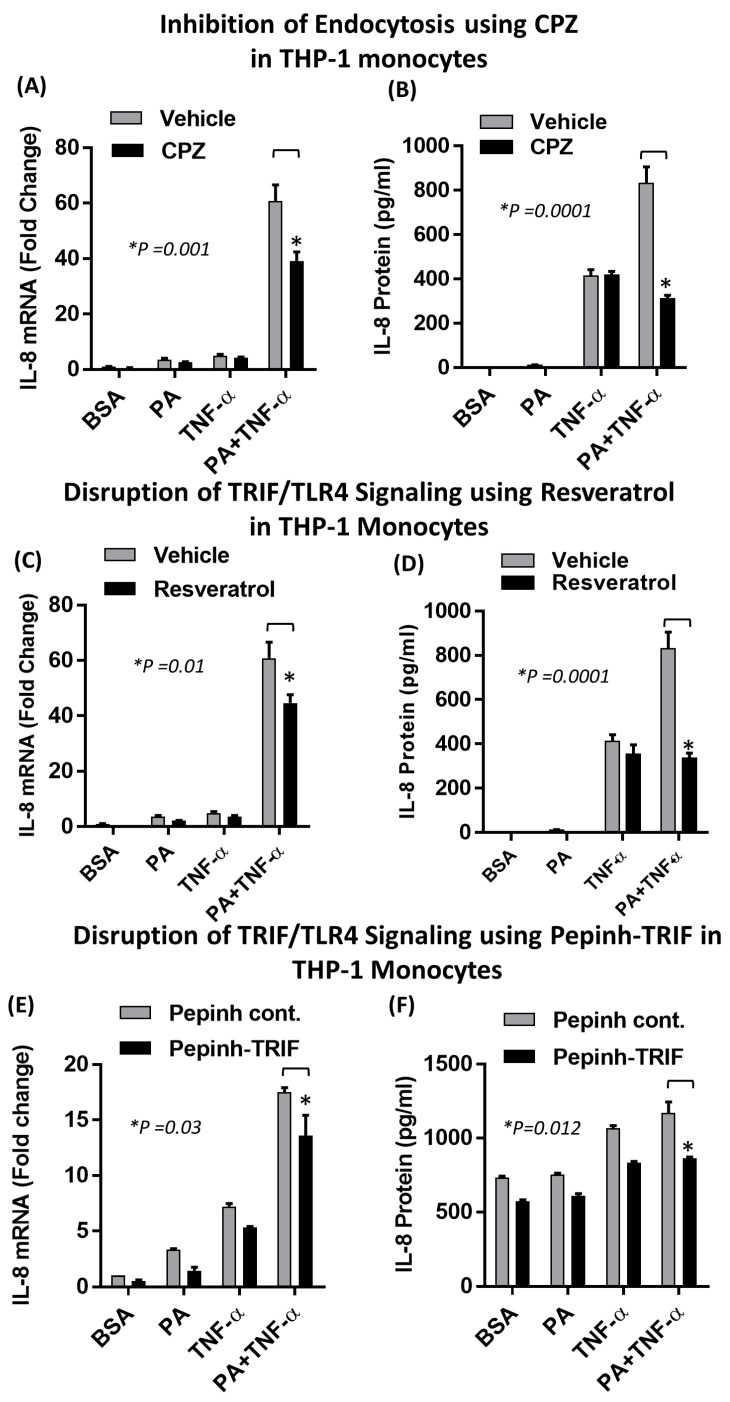
Synergistic IL-8 production by palmitate and TNF-α requires TRIF. (**A**,**B**) Monocytes were pretreated with chlorpromazine (chlorpromazine (CPZ); an inhibitor of endocytosis; 10 µM) for 1 h and then treated as indicated for 24 h. Cells were used for the isolation of total RNA to assess the *IL-8* gene expression by real-time reverse transcriptase polymerase chain reaction (RT-PCR). Cell culture supernatants were assayed for secreted IL-8 protein as quantified by ELISA. (**C**,**D**) Monocytes were treated with resveratrol (a TRIF inhibitor; 5uM) for 30 min followed by treatments as indicated. IL-8 expression was determined at both the mRNA and protein levels. (**E**,**F**) Monocytes were incubated with TRIF Inhibitory Peptide (Pepinh-TRIF is a 30 aa peptide that blocks TRIF signaling by interfering with TLR-TRIF interaction) Pepinh-TRIF: RQIKIWFQNRRMKWKK-FCEEFQVPGRGELH-NH2 (5 µM) and Control: RQIKIWFQNRRMKWKK-SLHGRGDPMEAFII-NH2 (5 uM) for 5h and then treated for 24h as indicated. IL-8 expression was determined at the mRNA and protein levels. The results obtained from three replicates of each experiment are shown. All data are expressed as mean ± SEM. * Indicates significant difference.

**Figure 7 ijms-20-04112-f007:**
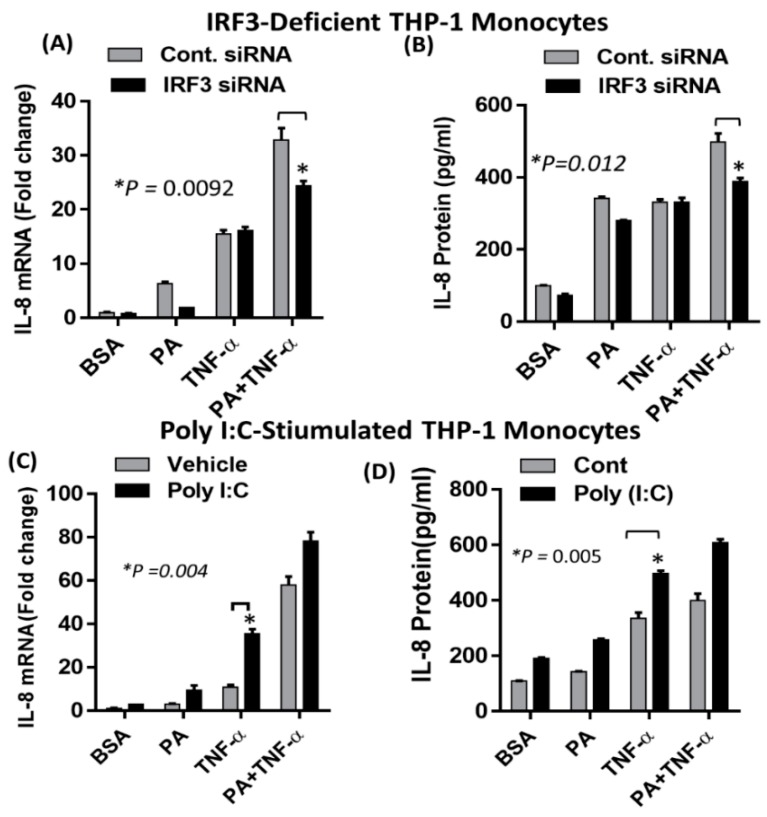
Synergistic production of IL-8 by palmitate/TNF-α requires IRF3. (**A**,**B**) Monocytes were transfected with IRF3 siRNA or control and incubated for 40 h. IRF3-deficient cells were treated with PA, TNF-α or a combination of PA and TNF-α. (**C**,**D**) Monocytes were treated (via transfection) with poly I:C (5 µg) for 2 h and then incubated with PA or TNF-α or combination of PA and TNF-α for 24 h. IL-8 mRNA expression was measured by real-time PCR and IL-8 protein was quantified by ELISA. The results obtained from three replicates of each experiment are shown. All data are expressed as mean ± SEM. * Indicates significant difference.

**Figure 8 ijms-20-04112-f008:**
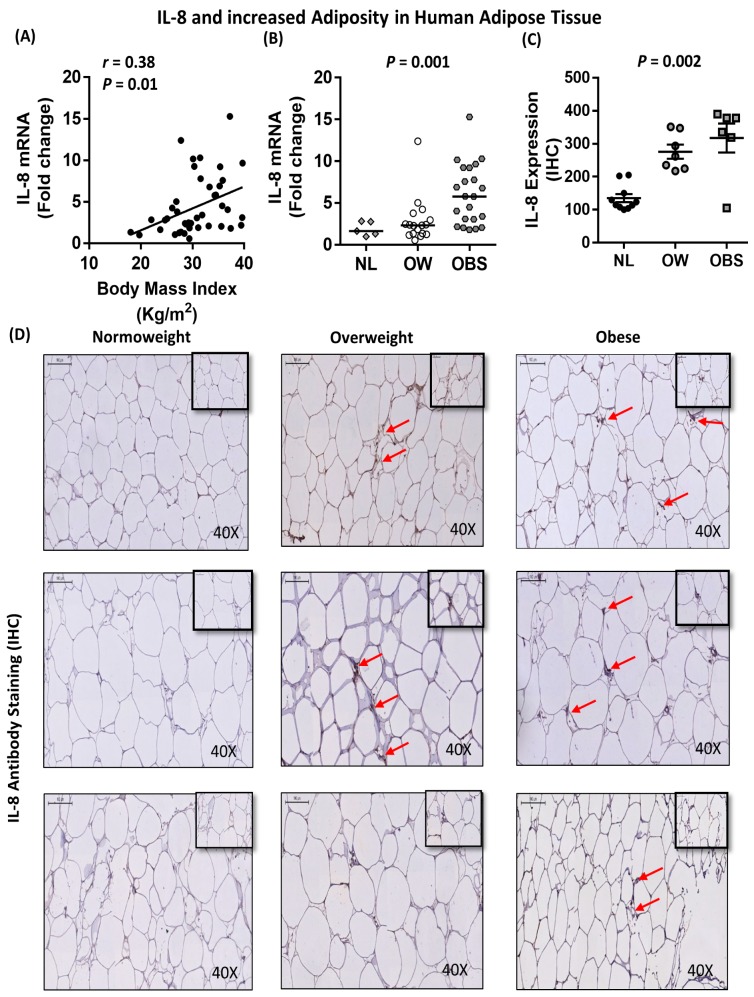
Association between IL-8 and increased adiposity in human adipose tissue. Subcutaneous adipose tissue was obtained from individuals with normoweight, overweight or obesity. (**A**) Direct correlation between adipose tissue IL-8 mRNA expression and the BMI. (**B**,**C**) Increased adipose tissue IL-8 mRNA and protein expression with increasing adiposity. IL-8 mRNA expression was detected by real-time RT-PCR and represented as fold change over controls. IL-8 protein expression was detected by immunohistochemistry and represented as staining intensity based on Aperio-positive pixel counts (Aperio software algorithm version 9.0). (**D**) Representative images are presented for IL-8 protein expression in the adipose tissue from individuals with normoweight, overweight or obesity. The arrowheads point to cells with intense staining for IL-8 protein. Correlation analysis was conducted using the non-parametric Spearman’s *r* test, and the group medians were compared using the non-parametric Kruskal-Wallis test.

**Figure 9 ijms-20-04112-f009:**
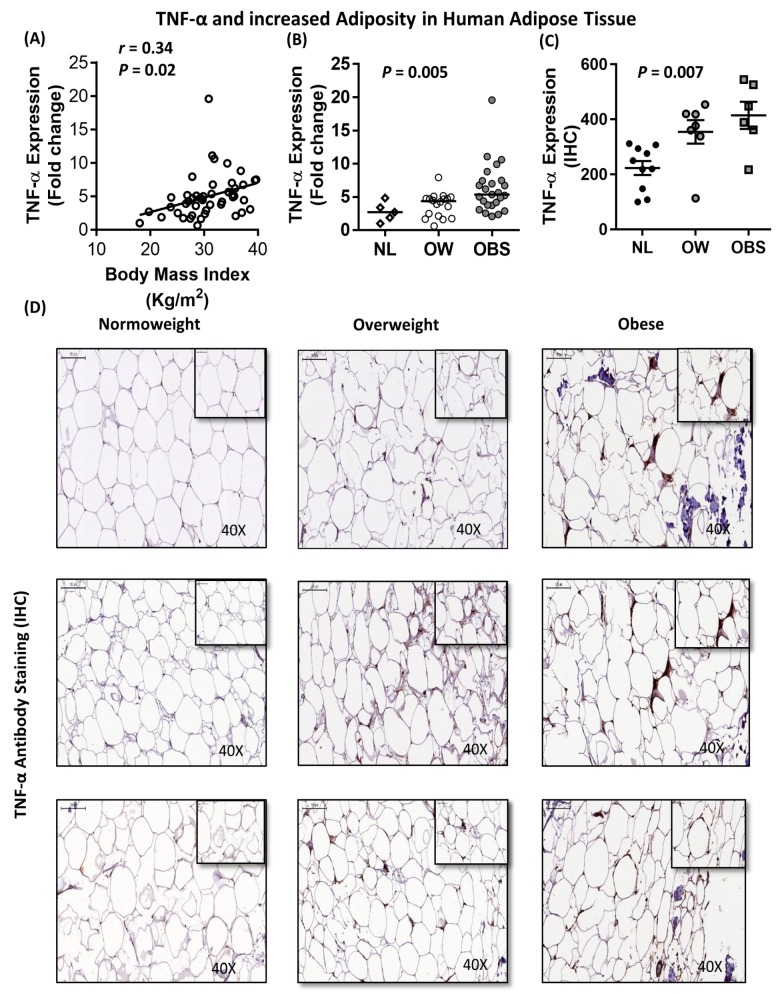
Association between TNF-α and increased adiposity in human adipose tissue. Subcutaneous adipose tissue was obtained from individuals with normoweight, overweight or obesity. (**A**) Direct correlation between adipose tissue TNF-α mRNA expression and the BMI. (**B**,**C**) Increased adipose tissue TNF-α mRNA and protein expression with increasing adiposity. TNF-α mRNA expression was detected by real-time RT-PCR and represented as fold change over controls. TNF-α protein expression was detected by immunohistochemistry and represented as staining intensity based on Aperio-positive pixel counts (Aperio software algorithm version 9.0). (**D**) Representative images are presented for TNF-α protein expression in the adipose tissue from individuals with normoweight, overweight or obesity. The arrowheads point to cells with intense staining for TNF-α protein. Correlation analysis was conducted using the non-parametric Spearman’s *r* test, and the group medians were compared using the non-parametric Kruskal-Wallis test.

**Figure 10 ijms-20-04112-f010:**
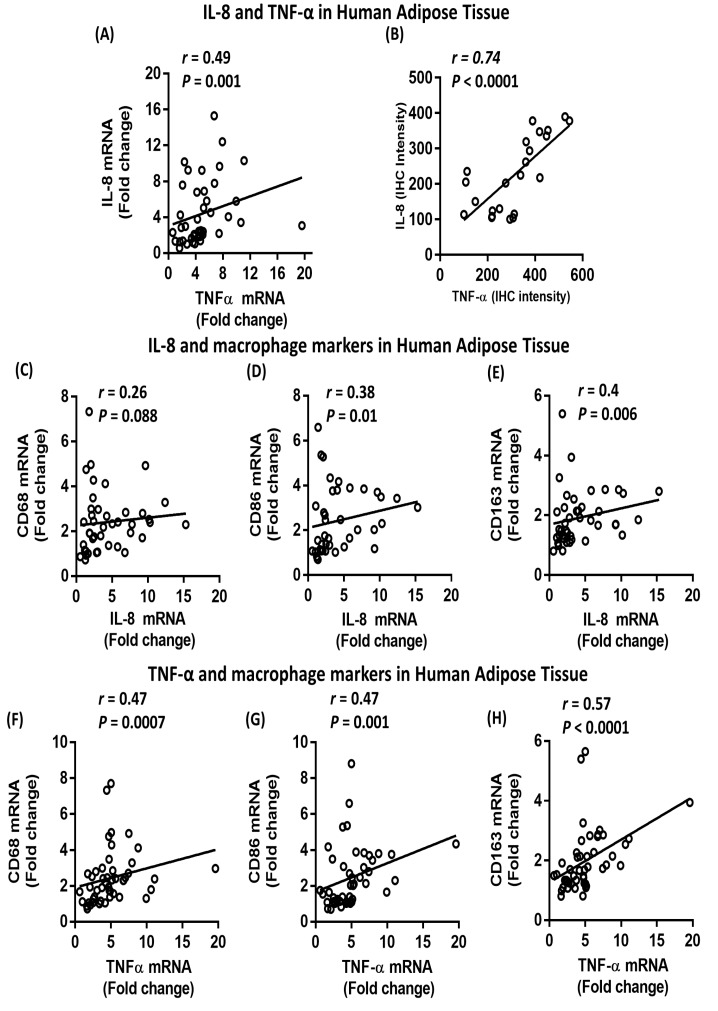
Association between IL-8, TNF-α and macrophage markers in human adipose tissue. Subcutaneous adipose tissue was obtained from individuals with normoweight, overweight or obesity. (**A**,**B**) Direct correlation between IL-8 and TNF-α at both the mRNA and protein level. (**C**–**E**) Direct correlation between IL-8 mRNA expression and CD68, CD86, and CD163. (**F**–**H**) Direct correlation between TNF-α and CD68, CD86 and CD163. The mRNA expression was detected by real-time RT-PCR and represented as fold change over controls. The protein expression was detected by immunohistochemistry and represented as staining intensity based on Aperio-positive pixel counts (Aperio software algorithm version 9.0). Correlation analysis was conducted using the non-parametric Spearman’s *r* test.

**Table 1 ijms-20-04112-t001:** Association of adipose tissue IL-8 with metabolic markers in humans.

Metabolic Markers	N	R	*P*-value
Body Fat (%)	38	0.39	0.016 *
BMI (kg/m^2^)	44	0.44	0.003 *
Fasting blood glucose (mmol/L)	44	0.04	0.79
Triglycerides (mmol/L)	44	0.16	0.29
Total cholesterol (mmol/L)	44	−0.02	0.89
HDL cholesterol (mmol/L)	44	−0.2	0.25
LDL cholesterol (mmol/L)	44	−0.001	0.99
HbA1c	43	0.15	0.33

* Indicates significant correlation.

**Table 2 ijms-20-04112-t002:** Anthropometric and biochemical characteristics of the study groups.

Participants	Normoweight	Overweight	Obese	* *P*-value
Total number (n)	13	18	27	-
Age (years)	43 ± 8.9	43 ± 10.7	44 ± 13	0.9
Body mass index (kg/m^2^)	23.4 (18, 24.7)	28 ± 1.36	34.7 ± 2.94	<0.0001 *
Percentage body fat (%)	26.87 ± 4.98 (n = 12)	33.2 ± 4.9	39.3 ± 4.04	<0.0001 *
Fasting glucose (mmol/L)	4.9 ± 0.67	5.12 ± 0.43	5.18 ± 0.52	0.45
† HbA1c (%)	5.6 ± 0.45	5.47 ± 0.32	5.7 (4.1, 6.2)	0.33
Total cholesterol (mmol/L)	5.2 ± 0.95	5.06 ± 0.8	5.04 ± 1.1	0.7
‡ HDL cholesterol (mmol/L)	1.56 ± 0.5	1.3 (0.84, 2.11)	1.18 ± 0.27	0.03*
§ LDL cholesterol (mmol/L)	3.3 ± 0.79	3.24 ± 0.7	3.33 ± 1.01	0.9
Triglyceride (mmol/L)	0.8 ± 0.4	1.08 ± 0.52	1.07 (0.42, 3.1)	0.1
¶ HsCRP (µg/mL)	2.05 ± 1.03	3.89 ± 2.7	5.5 ± 3.56	0.07
White blood cell count	5.8 ± 1.79 (n = 12)	5.8 ± 1.2	6.3 (3.8, 12.2)	0.7

**Data shown represent either the standard deviation of the mean or the median (min, max) as indicated.** * Non-parametric Kruskal–Wallis test of the medians. † Glycated haemoglobin. ‡ High-density lipoprotein. § Low-density lipoprotein. ¶ High-sensitivity c-reactive protein.

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
