# Peer review of "TNF-α in Combination with Palmitate Enhances IL-8 Production via The MyD88- Independent TLR4 Signaling Pathway: Potential Relevance to Metabolic Inflammation"

_ijms, 2019, doi:10.3390/ijms20174112_

Round 1

Reviewer 1 Report

This well conducted study demonstrates the synergistic pro-inflammatory effect of FFAs and TNF-a on the THP-1 monocytes and provides evidence for the signalling pathways involved. Besides in vitro data, the authors partly support their findings with human data.

In my opinion, the study needs to be improved and clarified in the following  parts:

the results of in vitro experiments on mRNA expression and protein levels should report the number of replicates;

THP-1 is an immortalized cell line. It should be of interest if the synergistic effect of TNF-a and FFAs on IL-8 should be reproduced in patients' monocytes and whether differences could be identified between the three investigated populations;

was the concentration of FFAs used in the experiments (200 uM) comparable with human data?

in table 1, the authors should specify the number of patients for whom adipose tissue was obtained; the patients' cohort did not show differences in glucose or HbA1c, and from biochemical data they all appear to be non diabetics. Moreover in results IL-8 was not correlated with glucose or HbA1c (table 2), while in the abstract the authors state that IL-8 correlated with these parameters (lines 26-27 page 1). In my opinion, patients' data  do not support the relationship between adipose tissue inflammation and diabetes, but only the association between obesity and inflammation. A large set of data is presented on patients, but their discussion is very limited;

page 14, line 292: please add the reference.

Author Response

Response to Reviewer 1 Comments

We appreciate the reviewer’s thoughtful comments which help us to improve our manuscript. We have now addressed his concerns in the revised version of our manuscript. Please see below point by point response to the comments made by reviewer.

Comments and Suggestions for Authors by Reviewer 1

This well conducted study demonstrates the synergistic pro-inflammatory effect of FFAs and TNF-a on the THP-1 monocytes and provides evidence for the signalling pathways involved. Besides in vitro data, the authors partly support their findings with human data.

In my opinion, the study needs to be improved and clarified in the following parts:

Comment: the results of in vitro experiments on mRNA expression and protein levels should report the number of replicates;

Author Response:  The results obtained from three replicates of each experiment are shown in the figures. Replicates related information was added in the figure legends as suggested.

Comment: THP-1 is an immortalized cell line. It should be of interest if the synergistic effect of TNF-a and FFAs on IL-8 should be reproduced in patients' monocytes and whether differences could be identified between the three investigated populations; was the concentration of FFAs used in the experiments (200 uM) comparable with human data?

Author Response: In the initial phase of study, we performed experiments with human primary monocytes, and they showed similar response to TNF-α and fatty acid palmitate for IL-8 production as we had seen in THP-1 monocytes. However, human primary monocytes show a synergistic response to palmitate for IL-8 production at very low (15µM) concentration when combined with TNF-α (100pg) (Supplemental Figure 1).

Comment: in table 1, the authors should specify the number of patients for whom adipose tissue was obtained; the patients' cohort did not show differences in glucose or HbA1c, and from biochemical data they all appear to be non-diabetics. Moreover, in results IL-8 was not correlated with glucose or HbA1c (table 2), while in the abstract the authors state that IL-8 correlated with these parameters (lines 26-27 page 1). In my opinion, patients' data do not support the relationship between adipose tissue inflammation and diabetes, but only the association between obesity and inflammation. A large set of data is presented on patients, but their discussion is very limited;

Author response: Corrections were made as per suggestion in the abstract.

Comment: page 14, line 292: please add the reference.

Author response: References have been added as suggested.

Reviewer 2 Report

This manuscript aimed to reveal the associations between palmitate, TNF-α, and IL-8 in human monocytic leukemia THP-1 cell line and human adipose tissue samples. The data demonstrated that the synergistic effect of palmitate and TNF-α on enhanced production of IL-8 was related to MyD88-independent pathway. Based on the validated approaches and suitable methodology, this manuscript deserves to be published. However, there are a few minor issues need to be addressed before it is suitable for publication.

“*P < 0.05” could be omitted from all figure legends. The productions of IL-8 protein in THP-1 cells by treating with palmitate alone, TNF-α alone, and PA+TNF-α in Figure 2 (A) were almost 3-fold of those in Figure 1 (B). Please explain the contradiction. Figure 4 (A) and line 164: What is Tsp000126? And where is JNK inhibitor (SP000125)? Line 167: The concentrations of Triptolide and resveratrol are missing from the legend of Figure 4. There is a typo in the subtitle of Figure 6 (E and F). It should be “Disruption of TRIF/TLR4 signaling…”.

Author Response

Response to Reviewer 2 Comments

We appreciate the reviewer’s thoughtful comments which help us to improve our manuscript. We have now addressed his concerns in the revised version of our manuscript. Please see below point by point response to the comments made by reviewer.

Comments and Suggestions for Authors by Reviewer 2

This manuscript aimed to reveal the associations between palmitate, TNF-α, and IL-8 in human monocytic leukemia THP-1 cell line and human adipose tissue samples. The data demonstrated that the synergistic effect of palmitate and TNF-α on enhanced production of IL-8 was related to MyD88-independent pathway. Based on the validated approaches and suitable methodology, this manuscript deserves to be published. However, there are a few minor issues need to be addressed before it is suitable for publication.

Comment: “*P < 0.05” could be omitted from all figure legends. The productions of IL-8 protein in THP-1 cells by treating with palmitate alone, TNF-α alone, and PA+TNF-α in Figure 2 (A) were almost 3-fold of those in Figure 1 (B). Please explain the contradiction. Figure 4 (A) and line 164: What is Tsp000126? And where is JNK inhibitor (SP000125)? Line 167: The concentrations of Triptolide and resveratrol are missing from the legend of Figure 4. There is a typo in the subtitle of Figure 6 (E and F). It should be “Disruption of TRIF/TLR4 signaling…”.

Author Response: “*P < 0.05” have been omitted from all figure legends.  We used THP-1 cells of different passages (n=3 -6) for our studies. All cell passages gave a similar synergistic response to TNF-α and palmitate for IL-8 production. In case of Fig. 2, the cells show higher base line response to Palmitate, TNF- α and similarly synergistic production of IL-8 in response to Palmitate/TNF-α. This base-line variation in the two figures could be due to different cell passages.  Tsp000126 should be SP000125 (JNK inhibitor), typing error has been corrected in the Figure 4 (A). The missing concentrations of triptolide (10µM) and resveratrol (5µM) have been added in the legend of Figure 4. Subtitle of Figure 6 (E and F) was corrected.

Reviewer 3 Report

In this study, the authors confirmed a synergistic activity of FFAs and TNF-α in IL-8 production by human monocytes in in vitro experiments. They further elucidated the synergistic effect between FFAs and TNF-α involved the MyD88-Independent TLR4 signaling pathway by using siRNA-mediated knockdown of TLR4 cells and MyD88 deficient cells. In addition, their human data showed direct correlation between the inflammatory activities in human adipose tissue and insulin resistance signatures. These findings provide a novel understanding of the pathophysiological mechanism of metabolic inflammation.

The manuscript was well written and easy to follow. The study was well conducted. Subject definitions and methods were described clearly. Tables and figures were used to present complicated information in a way that is accessible and understandable to the reader. The topic of this paper was relevant to the field of this journal. I have only one suggestion for improvement.

Line 241: Please delete ", and " in front of (Figure 10C, D and E)

I recommend acceptance of this paper after minor revision.

Author Response

Response to Reviewer 3 Comments

We appreciate the reviewer’s thoughtful comments which help us to improve our manuscript. We have now addressed his concerns in the revised version of our manuscript. Please see below response to the comment made by him.

Comments and Suggestions for Authors by reviewer 3

In this study, the authors confirmed a synergistic activity of FFAs and TNF-α in IL-8 production by human monocytes in in vitro experiments. They further elucidated the synergistic effect between FFAs and TNF-α involved the MyD88-Independent TLR4 signaling pathway by using siRNA-mediated knockdown of TLR4 cells and MyD88 deficient cells. In addition, their human data showed direct correlation between the inflammatory activities in human adipose tissue and insulin resistance signatures. These findings provide a novel understanding of the pathophysiological mechanism of metabolic inflammation.

The manuscript was well written and easy to follow. The study was well conducted. Subject definitions and methods were described clearly. Tables and figures were used to present complicated information in a way that is accessible and understandable to the reader. The topic of this paper was relevant to the field of this journal. I have only one suggestion for improvement.

Comment: Line 241: Please delete ", and " in front of (Figure 10C, D and E)

I recommend acceptance of this paper after minor revision.

 Author Response:

We made correction in Line 241 as suggested.  We deleted ", and " in front of (Figure 10C, D and E)